# A temporal classifier predicts histopathology state and parses acute-chronic phasing in inflammatory bowel disease patients

Lauren A. Peters [1], Joshua R. Friedman [2,3], Aleksandar Stojmirovic[3], Jacob Hagen[1], Sander Houten [1], Tetyana Dodatko[1], Mariana P. Amaro[1], Paula Restrepo[1], Zhi Chai [1], J. Rodrigo Mora[3,4], Holly A. Raymond[3], Mark Curran[3], Radu Dobrin[3,5], Anuk Das[3], Huabao Xiong [1], Eric E. Schadt [1], Carmen Argmann [1] & Bojan Losic [1,6]✉

Previous studies have conducted time course characterization of murine colitis models through transcriptional profiling of differential expression. We characterize the transcriptional landscape of acute and chronic models of dextran sodium sulfate (DSS) and adoptive transfer (AT) colitis to derive temporal gene expression and splicing signatures in blood and colonic tissue in order to capture dynamics of colitis remission and relapse. We identify sub networks of patient-derived causal networks that are enriched in these temporal signatures to distinguish acute and chronic disease components within the broader molecular landscape of IBD. The interaction between the DSS phenotype and chronological time-point naturally defines parsimonious temporal gene expression and splicing signatures associated with acute and chronic phases disease (as opposed to ordinary time-specific differential expression/splicing). We show these expression and splicing signatures are largely orthogonal, i.e. affect different genetic bodies, and that using machine learning, signatures are predictive of histopathological measures from both blood and intestinal data in murine colitis models as well as an independent cohort of IBD patients. Through access to longitudinal multi-scale profiling from disease tissue in IBD patient cohorts, we can apply this machine learning pipeline to generation of direct patient temporal multimodal regulatory signatures for prediction of histopathological outcomes.

[1] Department of Genetics and Genomic Sciences, Icahn School of Medicine at Mount Sinai, New York, NY, USA. [2] Spark Therapeutics, Philadelphia, PA, USA. [3] Janssen Research & Development, LLC, Spring House, Philadelphia, PA, USA. [4] Moderna, Cambridge, MA, USA. [5] Pathos AI, Berwyn, PA, USA. [6] Present address: Guardant Health, Redwood City, CA, USA. ✉email: blosic@gmail.com

nflammatory bowel disease (IBD) is characterized by a breakdown of intestinal immune homeostasis, mediated by cyclical flares of inflammation and driven by a complex interplay of genetics and environmental factors. Clinically, two major subsets of IBD are defined as Crohn's disease (CD) and Ulcerative colitis (UC) but their exact molecular etiologies remain elusive[1]. Murine models of colitis have provided some valuable insights into the molecular and histopathological changes associated with disease progression and therapeutic intervention. However, they have also been criticized for failing to fully recapitulate the complexity of CD and UC, as specific models depict different aspects of colitis development and progression. For example, the dextran sulfate sodium (DSS)-induced colitis mouse model induces colitis via barrier disruption and provides key insight into innate immunity-driven inflammation. On the other hand, adoptive T-cell transfer (AT) models represent T-cell-mediated colitis and can serve as a model for CD. Nonetheless, the DSS model has been successfully used to assess the effects of various perturbations, including therapeutic agents[2–4] and genetic perturbations[5–9], on both chronic and acute diseases.

Previous studies aimed at the characterization of the DSS model have focused on clinical[10], histopathological[10,11], and immunological[12–14] aspects of the disease, however, a multiscale model of the molecular alterations associated with disease dynamics demands genome-wide and systems approaches. In their earlier work, Breynaert et al.[15] developed a multi-DSS-cycle model mimicking the relapsing–remitting course of human colitis while exploring colonic gene expression alterations contrasting acute and chronic inflammation using microarray technology[15]. They identified distinct expression signatures for acute and chronic inflammation, which included overexpression of keratins associated with wound healing[15]. More recently, Holgersen et al.[16] performed RNA sequencing (RNAseq) on colonic tissue from IBD patients and three murine models of colitis: interleukin-10 (IL10) knockout, AT, and DSS[16]. They showed that, among the 115 genes previously identified in biomarker studies of IBD, 92 were differentially expressed in human colonic tissue and there was significant overlap with the genes deregulated in the Il10KO and adoptive transfer models[16]. They concluded that the colonic transcriptional profiles of human IBD and the murine colitis models are homologous[16]. We and others have also demonstrated the considerable utility of DSS models and their high resolution within human IBD ex vivo models as part of a comprehensive strategy for validating key driver genes[9,17].

The temporal dynamics of disease onset in colitis are of key importance, and are still largely uncharacterized, with clear potential applications in disease staging and potentially in developing treatment course prediction and management. In order to further parse the temporal dynamics of colitis and establish connections to key disease inflection points in IBD, we characterized the temporal variance of colitis-associated transcriptomic alterations in both acute and chronic mouse models. We characterize the disease-specific expression and splicing alterations across time and estimate their relative predictive power for histopathological measurements compared to other more traditional signatures. By projecting the most predictive signatures onto existing human IBD causal networks, we show that it is possible to explicitly identify acute and chronic subnetworks which elucidate potential transition points of acute to chronic colitis and thereby provide a quantitative framework of disease evolution.

## Results
### Repeated cycles of DSS exposure interspersed with recovery induce chronic disease pathology. We evaluated the induction

and evolution of DSS-induced colitis by measuring the primary endpoint phenotypes of this model: body weight, colon length, and histopathology scores. In line with disease model endpoints, mice in the disease group showed weight losses of 20% and 10% after DSS cycles one and three, respectively (Fig. 1a). No decrease in body weight was observed after DSS cycle two. All three DSS-free phases were accompanied by marked increases in body weight. The pronounced weight loss, shortening of the colon (Fig. 1b), and histopathology (Fig. 1c) confirm the development of a severe colitis phenotype. H&E staining of distal colon slices revealed edema and marked inflammation with moderate gland loss (Fig. 1d). No microscopic signs of recovery were observed on either day 12 or day 36. This suggests that the increased body weight observed in DSS-free phases one and three reflect a combination of recovery from systemic inflammation and incomplete colonic healing.

### Transcriptional features of DSS cycles. We performed RNAseq on intestine and whole blood samples collected on days 5, 12, 17, and 36. At each timepoint, we quantified gene expression and differential exon-usage patterns (cassette exon excision and retention), and carried out V(D)J alignment and CDR3 de novo assembly to quantify adaptive immune response magnitude and clonality (Supplementary Data 1, 2, Supplementary code 1). We used these features in a linear statistical model which tested for disease-specific differential expression signatures at each timepoint, which we called fixed colitis signatures, and also for disease-specific temporal variation of expression across all four timepoints (i.e., genes that significantly changed their differential expression profile across time), which we called dynamic colitis signatures.

### Fixed colitis expression and splicing signatures. The four timepoint-specific DSS vs. control differential gene expression signatures (DE) defined as logFC > 0 with FDR < 0.05 show an increase in transcriptomic alterations from day 5 (2040 DE genes) to day 17 (5740 DE genes), with a decrease in the levels of disease gene expression at day 36 (2771 DE genes) (Fig. 2a), suggesting an initial surge in molecular alterations followed by a recession and a disease-maintenance phase at later time-points. The complete table with logFC and FDR by DSS cycle, along with gene annotation, is provided in Supplementary Data 3.

The differential exon-usage signature, which we identify as the genes with significant evidence of differential splicing (DS), was found to be much weaker than gene-level expression changes in terms of number of genes involved at the same significance threshold. The highest number of significant DS genes peaked at day 12 with 125 genes having evidence of differential splicing at FDR < 0.05 (Fig. 2b; Supplementary Data 4). Both signals present high temporal specificity with 55% of DE genes and 82% of DS genes showing disease regulation specific to one-time point (Supplementary Fig. 1a, b). Although half of the DS genes (50.4%) are also differentially expressed at the gene level (Supplementary Fig. 1), expression and splicing alterations seem to be orthogonal to each other as first observed in Pan et al.[18], with the genes that show strong evidence for differential splicing having low gene-level expression fold changes and vice versa (Supplementary Fig. 2).

### Colitis in the mouse induces the expression of minor Il1rl1 and Lama3 isoforms. Among the 246 genes that showed significant evidence of being differentially spliced at any of the timepoints (Supplementary Fig. 1) we identified the genes that (a) have an FDR < $10^{-4}$ for differential splicing at least one timepoint and (b) whose differential exon-usage pattern can be explained by

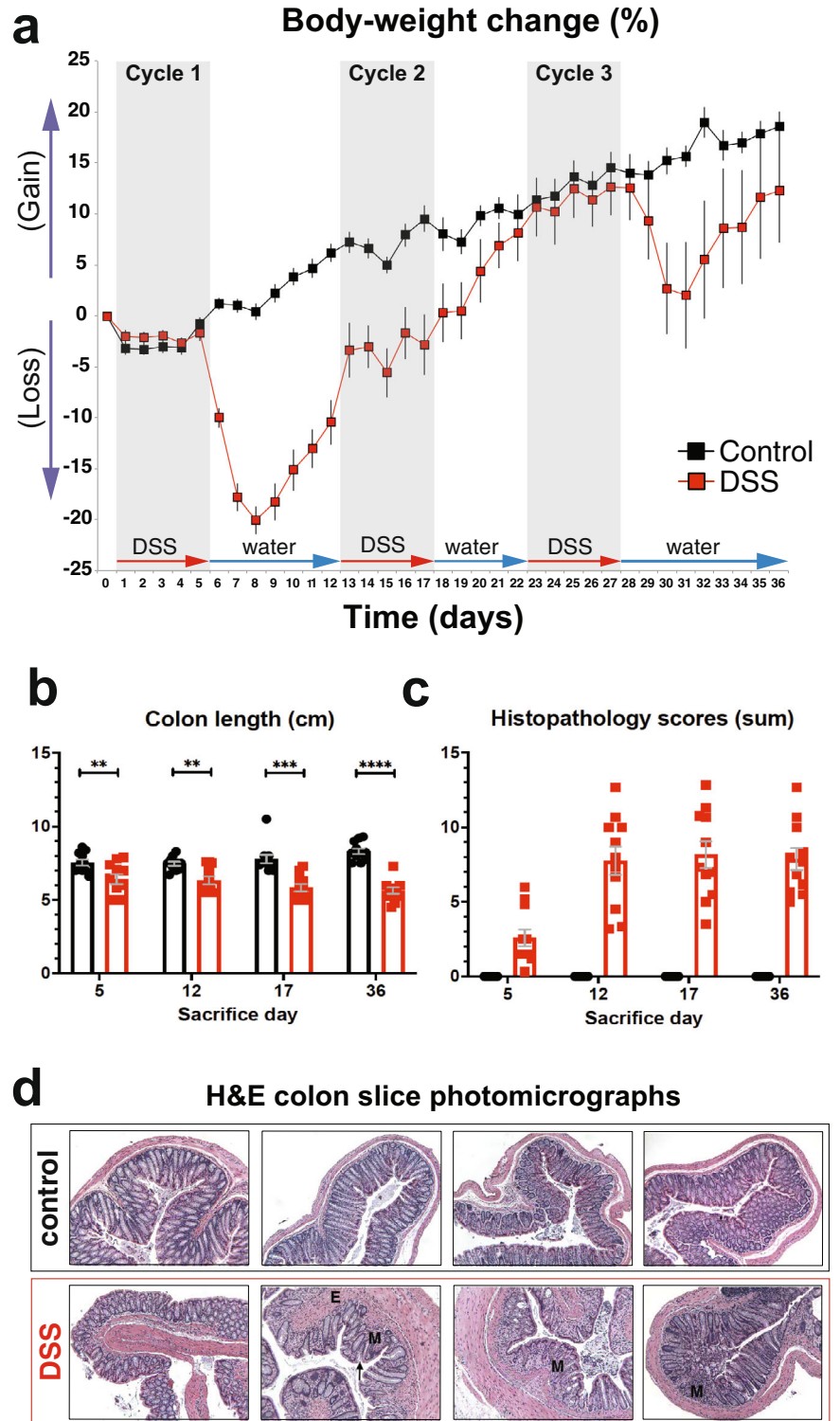

**Fig. 1 DSS-induced colitis phenotype. a–c** The evolution of colitis severity as exhibited by body-weight change (**a**), colon-length (**b**), and some of the histopathology scores (Wilcox *p* values) (**c**). Body weight was measured every day throughout the experiment; colon length and histopathology scores were measured at sacrifice days (5, 12, 17, and 34). Error bars denote one standard deviation across 6–24 mice per group. **d** Representative photomicrographs of H&E stained distal colon slices. Magnification: ×100. DSS colons show edema (E), severely affected areas of mucosa (M), and marked inflammation with moderate gland loss (arrow).

differential expression of annotated transcripts (in Refseq and/or Ensembl). Two genes met those criteria: Il1rl1 (ST2), a receptor in both membrane-bound and soluble forms belonging to the Toll-like receptor superfamily whose ligand is Il-33, and Lama3, a

secreted protein that belongs to the laminin family and acts as the alpha subunit of laminin-5 heterotrimer. The early DSS-specific Lama3 isoform has an FDR of 2.39e-18 (row 1 of Supplementary Data 6), while the early DSS-specific Il1rl1 isoform has an FDR of

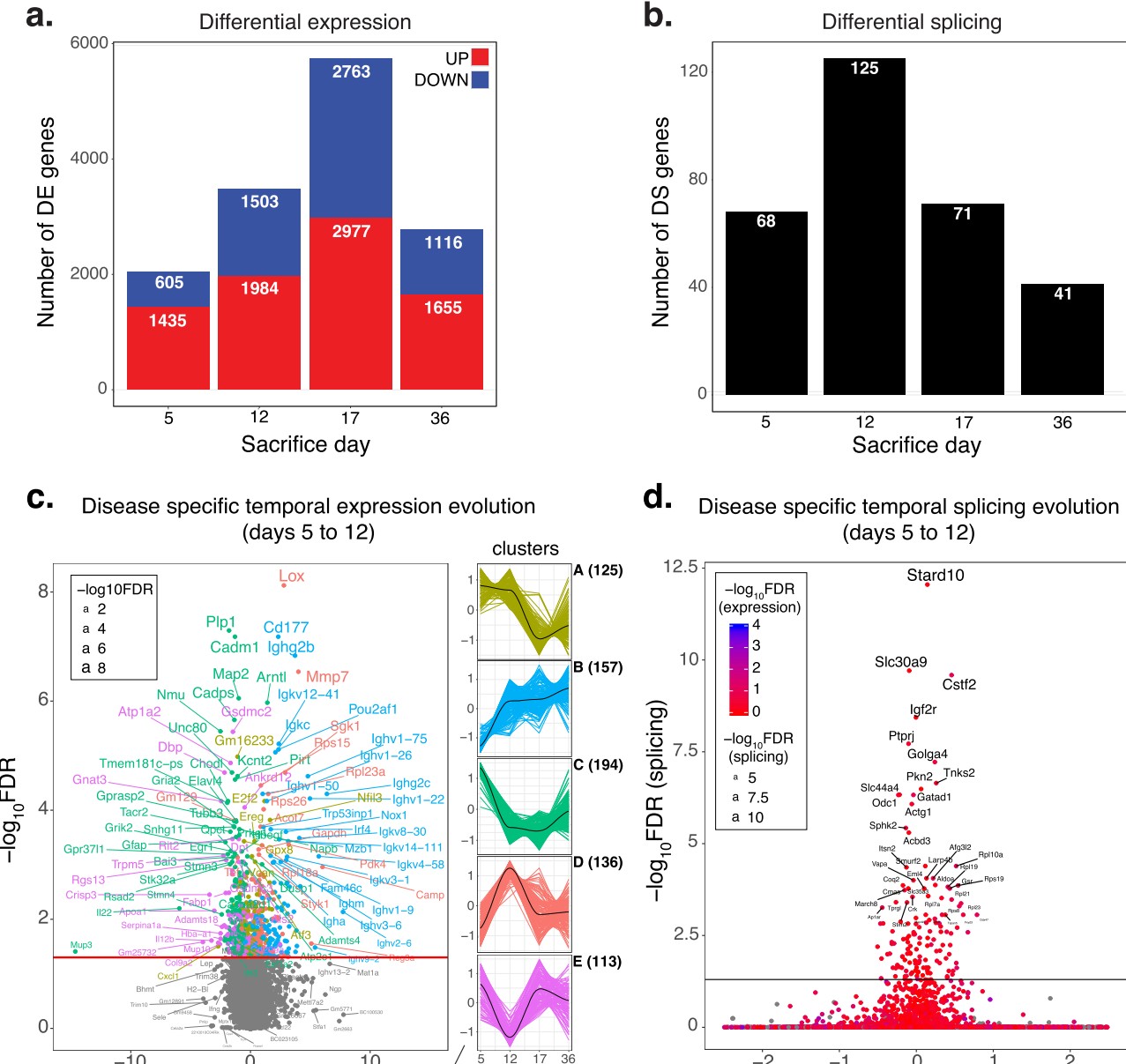

**Fig. 2 Temporal differential expression and splicing signatures and trajectories. a, b** Numbers of differentially expressed and spliced genes between control and disease at fixed sacrifice days (DSS hit/rest cycle). **c** Subset of differentially expressed genes with unique disease-specific temporal trajectories, which are clustered into five dominant classes each with 133–194 genes as indicated. Volcano plot depicting the change in log fold change between days 5 and 12 as a function of p value. Gene label colors match cluster membership. Horizontal line depicts 5% FDR threshold. **d** Orthogonality of disease-specific differential splicing and expression temporal profiles.

1.4e-e3. Both are differentially spliced on day 5 (first hit DSS cycle), while Lama3 is also differentially spliced on day 36.

A plot of the differential exon-usage results for Il1rl1 via visualization of the alignment of a selected control and a DSS sample as a sashimi plot (Fig. 3a) reveals that the differential exon-usage pattern observed comes from the overexpression of a short transcript (Refseq ID: NM_010743.3) in the disease group. That transcript codes for a short soluble isoform of Il1rl1 that lacks the intracellular Toll/interleukin-1 receptor (TIR) homology domain (Refseq ID: NP_034873.2). A qPCR experiment on the same samples using assays designed to specifically amplify the transcripts that code for the TIR-containing and TIR-lacking isoforms confirms the overexpression of the short transcript and the increase of the TIR-lacking/TIR-containing isoform ratio

from 1:1 to 3.5:1 (Fig. 3b, c). However, in a differential splicing analysis performed on an RNAseq dataset generated from biopsies taken across the intestine of IBD patients, we have not found any evidence of IL1RL1 being differentially spliced (Supplementary Data 5).

In order to elucidate the functional impact of the shift in Il1rl1 transcript usage associated with DSS colitis, we have computed isoform-specific expression from RNAseq data by measuring the counts associated with the exons specific to either isoform. We then combined the isoform-specific gene expression profile with the gene-level analysis data and selected the genes positively correlated with each of the isoform-specific genes above a threshold (Pearson's $R > 0.8$) and performed a pathway enrichment analysis (from WikiPathways, KEGG, and Reactome) on

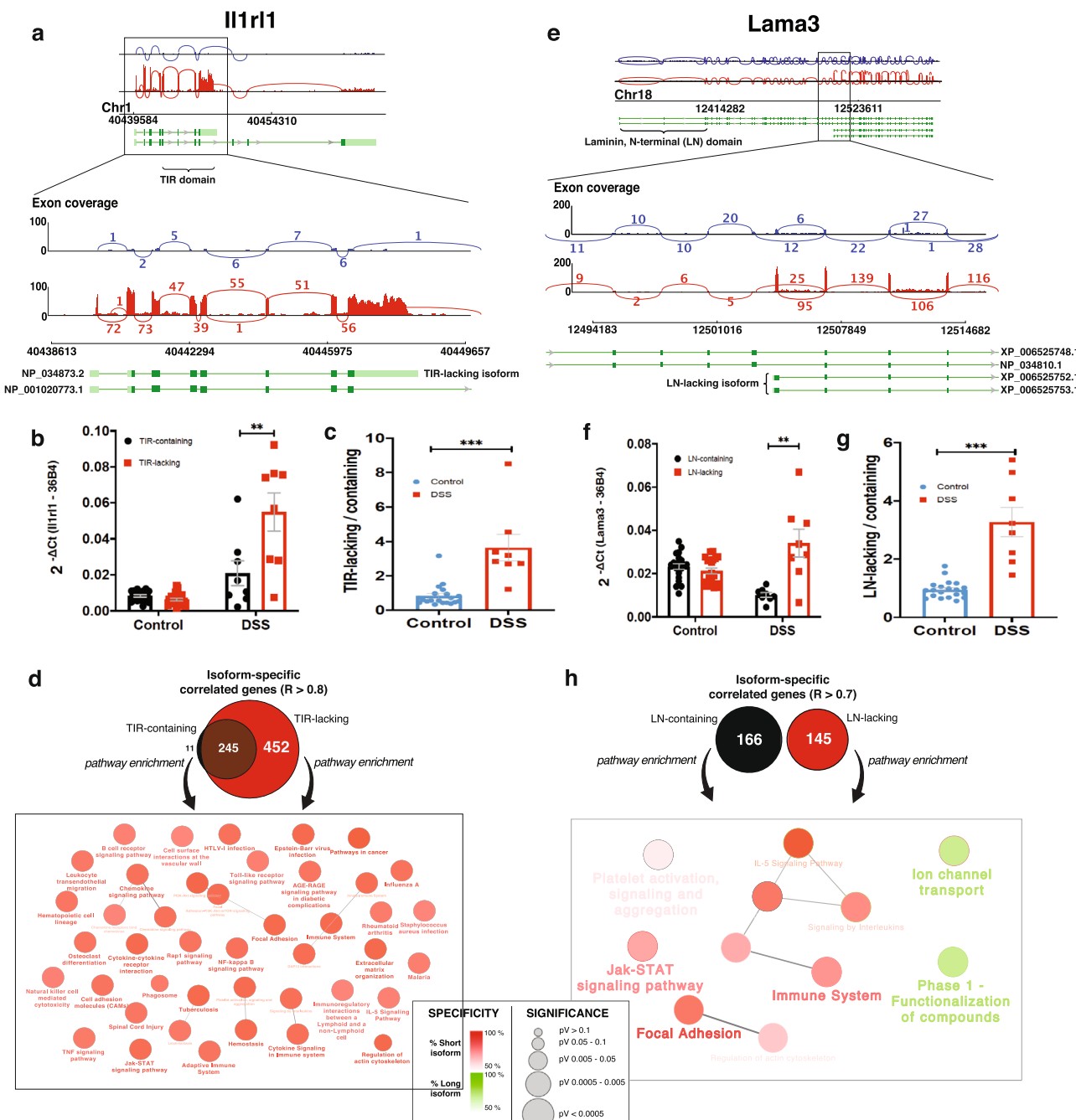

**Fig. 3 Differential splicing. a–d** Sashimi plot of Il1rl1 for two control mice (in blue) and two DSS mice (in red). Significantly varying exon and junction readings are shown in zoomed regions for Il1rl1 (**a**). DSS mice overexpress short transcripts of Il1rl1 that lack exons coding for the intracellular signaling Toll-interleukin-1 receptor (TIR) domain. **b** Relative expression of the TIR-containing and TIR-lacking Il1rl1 transcripts in the colon of the DSS and control groups given as RQ values (using 36B4 as endogenous control) obtained in qPCR validation. **c** Relative TIR-lacking/TIR-containing transcript expression calculated from the qPCR data. **e–h** Sashimi plot for Lama3 for two control mice (in blue) and two DSS mice (in red). Significantly varying exon and junction read support are shown in zoomed regions for Lama3 (**e**). DSS mice overexpress short transcripts of Lama3 that lack exons coding for the LN terminal (**f**, **g**), which have different functional enrichments (**h**).

the resulting isoform-specific co-correlated gene-lists. As the majority of the genes that are co-correlated with the long, TIR-containing isoform are also co-correlated with the short one (95.7%), none of the significantly enriched pathways are specific to the long-transcript (TIR-containing) gene list (Fig. 3d).

The differential exon-usage pattern for Lama3 (Fig. 3e) shows that a group of exons at the 5′ half of the gene are under-represented in disease. As with Il1rl1, the sashimi plot reveals that the differential exon-usage pattern arises from overexpression of a

short transcript (Refseq IDs: XM_006525689.3 or XM_006525690.2) in colitis. These transcripts code for a short Lama3 isoform (Refseq IDs: XP_006525752.1 or XP_006525753.1) that lacks the N-terminal laminin domain (LN-domain). Again, the results of a transcript-specific qPCR experiment confirm the overexpression of the short, LN-domain lacking isoform in colitis and show that it is accompanied by reduced expression of the long isoform, raising the short:long isoform expression ratio from 1:1 to 3:1 (Fig. 3f, g). Moreover, in the IBD patient dataset,

LAMA3 is differentially spliced (DS $p$val < .05), especially in the ileum, cecum, right colon, and rectum of CD patients, and in the cecum and rectum of UC patients (Supplementary Data 6). The results of the pathway enrichment analysis performed on the genes positively correlated (Pearson's $R > 0.7$) with either Lama3 transcript in RNAseq data show a shift towards immunity-related pathways, including Il-5 and Jak-STAT signaling, for the genes correlated with the short transcript (Fig. 3h).

**Dynamic gene, exon, and splicing expression signatures**. To identify genes that show different temporal expression profiles in the disease and control groups, we selected those for which the fit of the linear model was significantly improved when including a disease:time interaction term. At FDR < 0.05, 725 genes showed a disease-specific temporal expression pattern (Supplementary Data 7), with the majority (87%) being also differentially expressed in at least one time-point (Supplementary Fig. 1). By applying the soft-clustering Mfuzz algorithm (Kumar & Futschik, Bioinformation, 2007) on the median disease trajectories, we identified five clusters of genes: 1) early genes (cluster A, high expression at days 5 and 12; 113 genes), 2) late genes (cluster B, progressively increasing expression; 125 genes), 3) very early genes (cluster C, highest expression at day 5; 194 genes), 4) day 12 up-regulated genes (cluster D, highest expression at day 12; 157 genes), and 5) day 12 down-regulated genes (cluster E, lowest expression at day 12; 136 genes) (Fig. 2c). Among these, cluster B was strongly enriched in immunoglobulin genes (Supplementary Fig. 3). The expression of the members of this cluster shows a continuous increase from day 5 to day 36, pointing at a gradually stronger involvement of B cells in the pathogenic process in chronic disease. Taken together, these genes represent a small subset of differentially expressed genes at specific times that also differentially vary in time in diseased tissue. This smaller set of genes reflects dynamic disease evolution which is ordinarily averaged out in a case-control experimental design.

To identify genes with disease-specific splicing patterns across timepoints, a model containing a disease:time interaction term was fitted on exon-level data and the genes that showed evidence of differential change in exon-usage across timepoints were identified with a simple test. At FDR < 5%, 141 genes showed significant evidence for differential evolution of splicing in the DSS and control groups (Supplementary Data 8). Among those, 51 (36%) were differentially spliced at any single time-point (Supplementary Fig. 1). Even more pronounced than for the timepoint-specific results, the alterations observed in temporal trajectories for expression and splicing provide orthogonal information, as only 2 of the genes show different trajectories in disease and control for both splicing and expression (Supplementary Fig. 1) and the genes that show strong evidence for differential splicing evolution show the weak signal of disease-specificity of temporal expression patterns (Fig. 2d).

**Tracing adaptive inflammatory processes throughout time via adaptive immune profiling**. To further elucidate the magnitude and specificity of adaptive response throughout distinct disease phases, we quantified the B and T cell receptor (B/TCR) RNAseq reads mapping to VDJ loci, normalized by total library size. The non-targeted and sparse nature of these data prevents a deep characterization of the lymphocyte receptor repertoire which is directly comparable to that of targeted amplicon DNA-based sequencing[19]. Nevertheless, we and others have previously shown that RNAseq data can be used to infer reliable estimates of dominant immune clonotypes as a reasonable proxy of immune clonality in the context of inflammation[20].

Applying this technique to our RNAseq dataset across DSS hit-rest cycles, we see a clear correlation between increasing pathological lymphocyte aggregate counts in colonic tissue in the first DSS phase and VDJ measurements in the same (spearman rho ~0.86, $p$val ~0.05). Clonal diversity was reduced in later rest cycles. VDJ clones were detected in >99% of the DSS models at a median of 500 clones per sample, while on average 44% of samples in the adoptive T cell models had detections of only about 8 clones per sample (Supplementary Figs. 3, 4). We use this detailed molecular characterization of the time-dependent magnitude and clonality of the adaptive response as a key metric of colonic immune response.

**Testing the predictive power of expression and splicing signatures in IBD**. In order to assess the predictive power of the cycling and fixed signatures derived from the whole-colon Janssen DSS cohort, we used fixed and trajectory expression and splicing as features to predict histological scoring data across different: experimental lab sites (Janssen and MSSM), tissues (blood and colonic), and colitis model type (DSS and AT)). For each signature, we constructed a random forest model using the same DSS whole-colon training set to predict histological scoring that incorporated predictive terms related to the specific fixed and trajectory expression, splicing, and adaptive VDJ signatures evaluated on the validation data (see Methods, Supplementary Fig. 5). As shown in Fig. 4, we ultimately tested the performance, and feature importance, of multiple signatures on multiple validation sets. Since our objective variable, histological scoring, is continuous we assessed performance using the spearman correlation and its associated $p$ value. By tuning each random forest model via 10-fold cross-validation, we ensured that only the most important molecular features were retained for each model even though initially each model had the same number of predictors distributed among expression, splicing, and VDJ features. As summarized in Fig. 4a, we used each specific fixed timepoint signature (5 vs 5), cycling differential expression signature, and splicing signature to predict histology scoring across different validation sets with the most predictive features summarized. These model features similarly predicted the averaged (all day) signature set (20 vs 20). The validation sets included the DSS Janssen Blood dataset, which has the highest risk of being batch-correlated with the DSS whole-colon signatures, the DSS MSSM blood and MSSM proximal colon (PC)/distal colon (DC) dataset, and finally the two AT models (TC1 and TC2) with both colon and blood expression. The distribution of histology scoring among the AT and DSS models and experimental lab sites is shown in Fig. 4b, underscoring the difference between the relatively uniform frequency of low and high histological damage in DSS models and that of AT models which are heavily peaked around low to moderate histological damage.

Our results suggest that the dynamic trajectory expression signature D is the most predictive of histology scoring in the DSS proximal colon data (rho ~0.8, -log10(p) ~8), and that this predictive power depends mostly on the gene expression features, not on adaptive inflammation or exonic usage. Similarly, dynamic trajectory signature B (immune influx) has moderate predictive power in DSS blood (rho ~ 0.5, -log10(p) ~3) again with expression features dominating the model. For the adoptive transfer model, we see that the DSS day 36 average control vs disease fixed signature is predictive for histological scoring using distal mucosal expression, and notably highlights the predictive power of differential exon usage (alternative splicing). Finally, averaging all DSS day-specific signatures into one, we observe a negative correlation between histology predicted in the adoptive transfer model using proximal mucosal expression, supporting

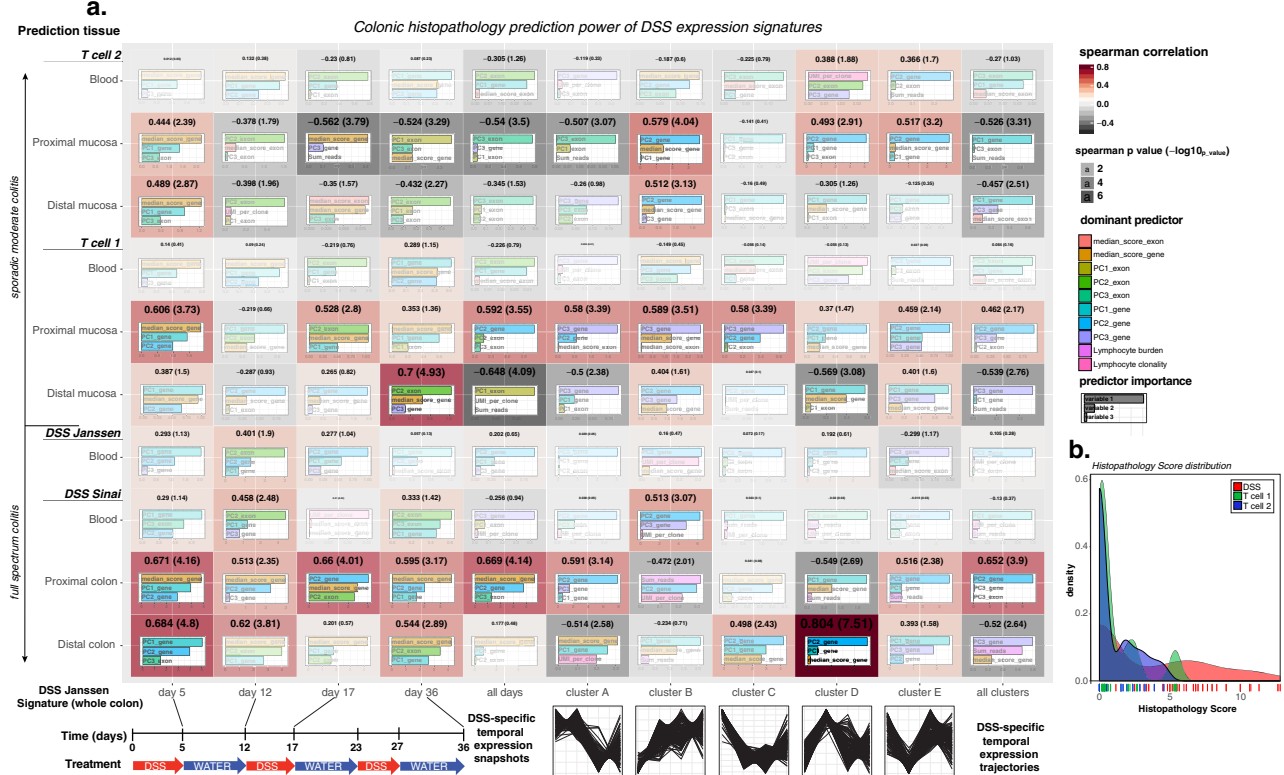

**Fig. 4 DSS temporal signature and trajectory predictive power of colonic histopathology across tissue, location, and colitis model.** The training data comprise DSS whole-colon temporal expression/splicing signatures and trajectories and V(D)J clonal deconvolution. The signatures are dimensionally reduced to the first three principal components and median expression score. A random forest model is trained on all of the DSS WC Janssen samples. Validation of the trained model was performed over different tissues, DSS experimental locations, and adoptive transfer colitis models to predict colonic histological scoring (or disease status). Prediction efficacy was evaluated via Spearman correlation value (and p value) and a corresponding Kappa value computed for disease status classification. DSS temporal signature and trajectory predictive power of colonic histopathology across tissue, location, and colitis model. **a** Heatmap of spearman correlation values (with value as color, significance as opacity and size) between predicted and observed histopathology. The relative predictor importance to each predictive model is indicated by the bar-charts. **b** Distribution of histopathology scores across models. Over 36 days the DSS models induce higher disease severity than either T-cell model.

that these models manifest disease in different colonic regions and time course.

The murine model molecular signatures were then tested for their ability to predict the Nancy (UC) and GHAS (CD) histological scores, as well as other related scores, in human IBD patient biopsies (Fig. 5) across different regions in an independent cohort of 1000 IBD patients (MSCCR cohort) Supplementary Data 9)). Specifically, we demonstrate the predictive power of the homologs of the colitis evolution signatures in predicting biopsy-region-averaged histological scores and subscores in Fig. 5. Exactly as in the evaluation of the predictive power of the DSS models themselves, each signature was broken down into the first several principal components (PC1–PC4) (Supplementary Fig. 6) and these were used in the prediction models.

**Translating the modulated transcriptomic features in the mouse DSS model to human IBD networks to further inform on human IBD pathology.** The relevance of the DSS-associated transcriptomic alterations to human disease was assessed by overlapping the murine DSS signatures obtained in this study to a causal gene-regulatory network (GRN) built from gene expression and genotyping data from intestinal biopsies of anti TNFα CD patients undergoing a Phase II trial for Ustekinumab[9]. The overlap of the timepoint-specific DSS signatures with the human GRN yields four subnetworks that show considerable overlap, with 52% of the genes appearing in two or more

subnetworks (Fig. 6a). In agreement with the degree of overlap between the sub- networks, a KEGG pathway enrichment analysis shows a large number of pathways (28) enriched in two or more time points, with three pathways (Staphylococcus aureus infection, cytokine-cytokine receptor interaction and leukocyte transendothelial migration) showing enrichment across all subnetworks (Fig. 6b). The two extreme temporal points share pathway enrichments absent at intermediate times, including the TNFα and NF-kappa B signaling pathways. This suggests that some of the immune alterations that trigger the pathogenic process at the beginning and that taper down at intermediate times are reactivated with repeated cycles of DSS treatment (Fig. 6b).

Some of the altered pathways involved in colitis have timepoint-specific enrichment. While day 5 shows enrichment for innate immune (phagosome), anti-viral response (Influenza A, HTLV- I infection) and mucosal immunoglobulin-mediated response pathways (intestinal immune network for IgA production), day 36 is specifically enriched for adaptive/MHC-mediated immune pathways (T cell receptor signaling, natural killer cell-mediated cytotoxicity, graft-versus-host disease) and genes that have been implicated in IBD pathogenesis (Fig. 6b). These results, along with the observation of the progressive increase in the expression of immunoglobulin genes (Supplementary Fig. 7), suggest that the resident immune response mechanisms to bacterial invasion are active within few days (they are active at day 5), while the systemic adaptive response that relies on the

## Tissue temporal signatures predict histological scoring, by contrast to blood

**a.**

**b.**

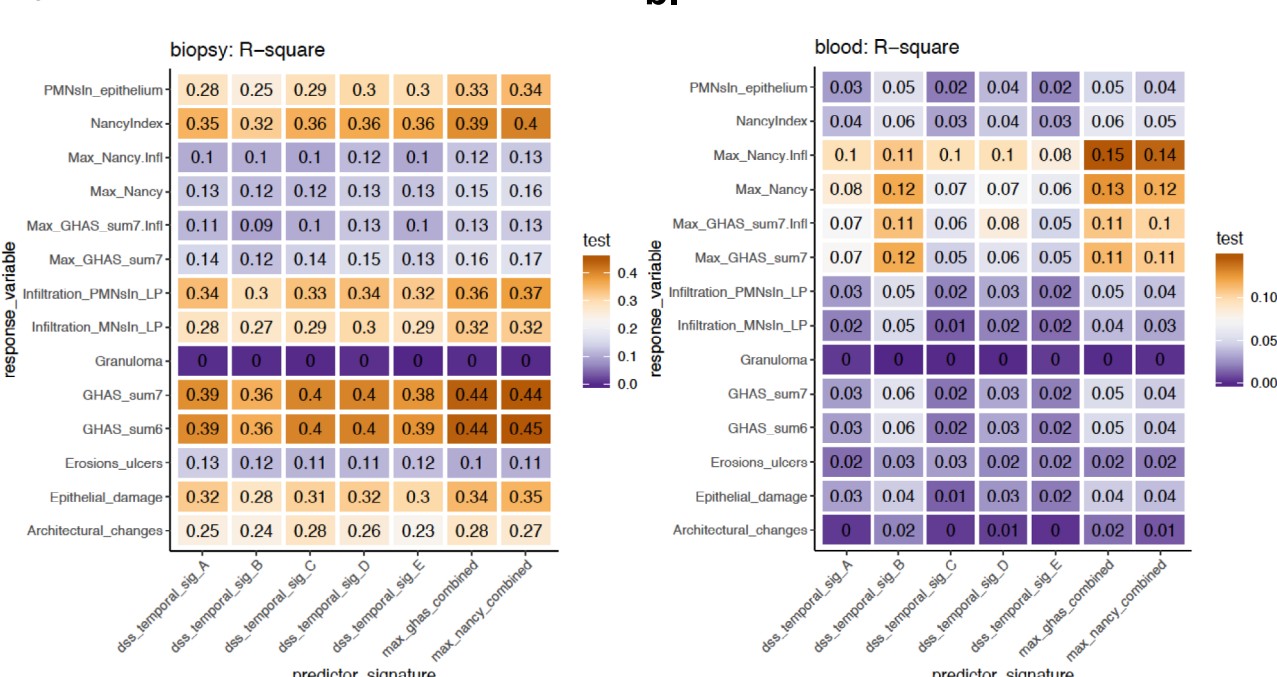

**Fig. 5 Mouse temporal signatures are predictive of histology state in a cohort of inflammatory bowel disease patients.** Comparison of adjusted R squared values for penalized ordinary least squares models of histological measurements using human orthologs of DSS disease evolution signatures and combined Nancy (UC) and GHAS (CD) signatures in **a**) (non-Ileal) patient tissue samples and **b**) patient blood samples. Each model fit the first five principal components of each signature (including splines with three knots to test for non-linear associations), penalized by extremizing the Akaike Information Criterion (AIC), and bias-corrected R-squared values were computed via 300 rounds of bootstrap resampling with replacement. Tissue specific histological scores per patient were averaged across multi-regional tissue samples.

presentation of antigens by the MHC molecules is not fully developed until 2–3 weeks (Fig. 2c and Supplementary Fig. 3a).

As with the pathways, days 5 and 36 show the strongest enrichment in genes of three IBD relevant gene-sets: an intestinal inflammatory signature of CD patients (CD signature), the list of candidate genes identified in IBD GWAS studies (IBD genes) and the genes involved in very early onset IBD (veoIBD genes) (Fig. 6b). While the day 5 subnetwork is more strongly enriched in genes of the intestinal inflammatory signature, the day 36 subnetwork shows stronger enrichment for genes potentially involved in the genetic etiology of the disease, resolving in the networks both the acute and chronic disease programs of relapse remitting inflammation, which we have shown are predictive of histological pathology.

### Discussion

We have used integrative analysis of RNAseq expression profiling of the blood and intestine to determine the degrees to which two experimental murine models of colitis relate to human IBD. We leveraged the power of RNAseq to characterize the differential splicing induced by disease, which highlighted isoform-specific contributions of the Lama3 and Il1rl1 loci. We then evaluated the predictive power of differential expression, differential splicing, VDJ repertoire and temporal clusters spanning acute and chronic phases within the same tissue and across tissue, disease model and species, for histological measurements. The temporal signatures had the most predictive power and integrating these regulatory readouts over time provided greater predictive power over single timepoints. We have also shown that human causal IBD networks can be intersected with the longitudinal murine data to parse acute versus chronic stages and putative transition

phases. We were also able to highlight the conservation of intermediate molecular phenotypes as the murine trajectory signatures overlapped with the transcriptional causal networks derived from the intestine from IBD patient populations. We have also shown that the human causal IBD networks can be intersected with the longitudinal murine data to parse acute versus chronic stages and putative transition phases.

The predictive power of these colitis model signatures extends to histological scores in a cohort of IBD patients. We used GHAS max as an outcome measure because the area of worst disease/ inflammation determines the score. However, there could be limitations with averaging and regressing out regional effects as intestinal transcription is known to vary in a region dependent manner, even with the exclusion of ileum. While the sample representation was outnumbered in rectum, leading to a dominance in transcriptional signal, regressing out region may affect biology of disease signal as it is thought that there could be regional differences in disease activity. Another limitation is that averaging intestinal tissues across regions may be dilutive to the expression of true disease signal, if there is a preponderance of noninflamed affected regions included. However, this analysis can also capture sub clinical molecular changes on a pathological escalation. Furthermore, capturing transcriptional predictive power in CD will differ from UC in disease region since UC is continuous progression distal to proximal whereas and CD is sporadic in its regional intestinal manifestation.

Nevertheless, this time series predictive pipeline has significant potential in being applied to large human datasets where longitudinal molecular profiling data is available from intestinal biopsy and/or blood. It may be used to generate IBD patient sourced temporal trajectories for the prediction of disease

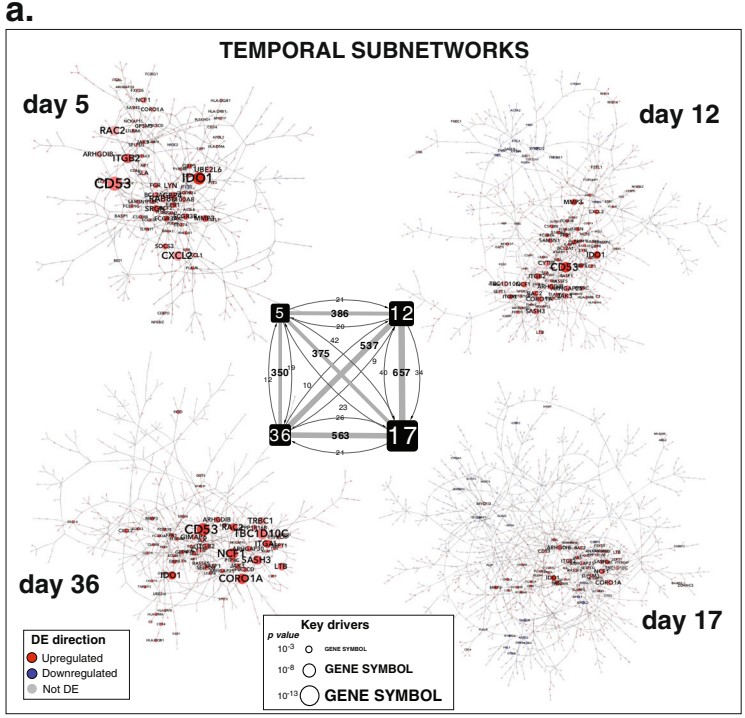

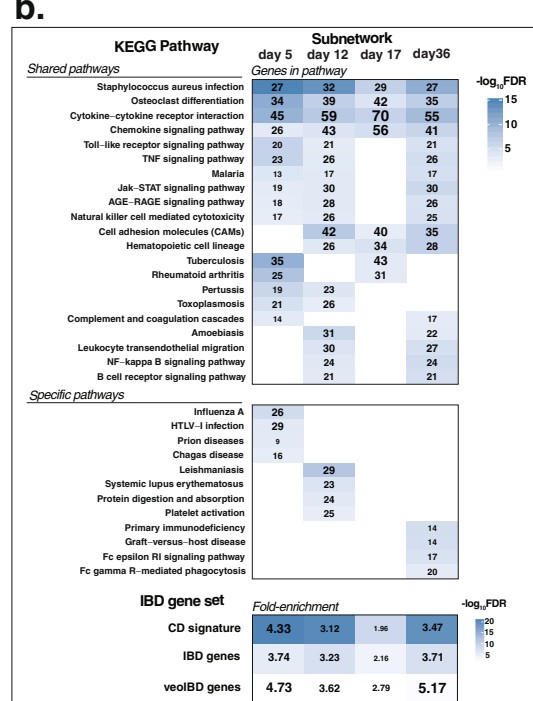

**Fig. 6 DSS temporal signature projection onto a human intestinal IBD causal network.** Colonic DSS temporal disease signatures were projected onto a Bayesian gene-regulatory network built from genotype and gene expression data of colonic samples from patients with Crohn's disease. **a** The 862 colitis-genes that mapped show statistically significant overlap with the 905 genes of a subnetwork previously associated to Crohn's disease (263 common genes, −log(p) = 44). The visualized subnetwork shows these common genes, along with their first neighbors (undirected, 959 genes in total). The blue-white-red color scale reflects the disease vs. control logFC of the genes in the mouse model. Gray genes were not differentially expressed in colitis. **b** Gene-set enrichment analysis (GSEA) was performed on both the mouse signature and the human subnetwork genes for enrichment in hallmark signatures of the Molecular Signature Database (MSigDB). Among the unique 42 molecular signatures found to be enriched in any of the gene-lists, 32 are shared, 7 are mouse signature-specific and 3 human subnetwork-specific. The FDR values of enrichment for mouse and human can be seen in the scatterplot, where the size of the circles represents the average number of genes in the overlap and the blue color gradient reflects the average −log (FDR).

| Table 1 DSS protocol. | | | | | | | |
|---|---|---|---|---|---|---|---|
| **Days** | | **1–5** | **6–12** | **13–17** | **18–23** | **24–27** | **28–34** |
| Group 1 | $N = 90$ | DSS | Water | DSS | Water | DSS | Water |
| Group 2 | $N = 90$ | Water | Water | Water | Water | Water | Water |

severity, mucosal healing and therapeutic response and could be of particular value if non-invasive tissue, such as the blood, can be used to predict local disease status in the intestine. This study highlights the utility of profiling aspects of experimental animal disease models and will facilitate more precise use of the models to test candidate therapies. Furthermore, this machine learning pipeline can be adjusted to generate expression or other data regulatory scale based histological annotation in convolutional neural networks to better classify the molecular signatures of histological stages and pathological tissue changes and identify non-linear predictive features. Further refinements of this model may enable the classification of patient subsets based on temporal signatures of disease if more routine longitudinal patient molecular profiling can facilitate a personalized medicine approach where a patient remission and flare can be better predicted in addition to defining sequential and/or combinatorial therapies tailored to specific intervention time points. As more longitudinal molecular data becomes available at various data scales, it can be paired with clinical data in order to annotate and track variables which may be driving the temporal clusters. Based on the conserved homology, various perturbations can be tested in experimental models and then profiled and re-integrated with human

data models leveraging longitudinal sample collection and profiling in order to better predict efficacy of in silico interventions for human clinical trials.

## Methods

**Murine colitis experiments (at Mount Sinai site): DSS.** On Day 0, female C57Bl/6 mice were weighed and divided into 2 groups of 90 mice per group such that the average weight of each group is similar. Mice were enrolled onto study at body weight of 16–18 grams. Enrolling mice at a range of 16–18 grams ensures consistent disease across the groups while minimizing losses due to death. Mice in Group 1 received 2.5% DSS in drinking water at times shown below while mice in group 2 received plain drinking water at all times. Food was provided ad libitum. See Table 1 below.

**Clinical assessment.** Mice were weighed daily for the duration of the study.

**Termination.** At various times during the study (Day 5, 12, 17, 23, 27, and 34) 15 mice from each group were chosen randomly and sacrificed after an overnight fast. Colon was excised and feces were harvested aseptically and placed into cryotubes and frozen. After feces collection, all contents of the colon were removed and colons flushed with cold PBS. Feces collection only occurred on days 5, 17, and 34. For all mice of each group, the 1 cm distal adjacent to the rectum was sectioned and placed in 10% buffered formalin for subsequent histological analysis. The colons from 15 mice per group at each time point were collected for RNA analysis. 1 cm distal adjacent to rectum was used for histology and the remaining tissue was used for RNA. The remaining colon tissues were frozen in RNAlater at −80°C for RNA extraction. Blood was collected by cardiac puncture from mice from every group at termination. Approximately 250 μl of whole blood was collected into PAXgene tubes. Whole blood was assessed for RNA. Colons from all groups were measured for length and weight. RNA was isolated with Qiagen RNeasy kits according to manufacturer's instructions.

**Adoptive transfer**. In this model, CD45RBhigh CD4+ T cells from C57Bl/6 J mice were transferred to RAG1−/− mice which lack T and B cells.

**Induction**. On study day 0, spleens for CD45RBhigh CD4+ T-cell isolation were obtained from female C57Bl/6 J mice using the RAG1−/− mice as recipient mice. After cells had been obtained and sorted, each female RAG1−/− mouse received an IP injection of $5 \times 10^5$ cells/ml (100 μl/mouse injection). Control RAG1−/− mice served as a control.

**Clinical assessment**. Mice were weighed 2–3 times a week for the duration of the study.

**Termination**. At various times during the study (Day 14, 21, 28, 35, and 42), 10 mice from each group were chosen randomly and sacrificed after an overnight fast. Large bowel was excised and contents flushed with cold PBS. The 1 cm part of distal colon was sectioned and placed in 10% buffered formalin for subsequent histological analysis. The remaining part of colon from the 10 mice, per group, per time point, were collected for RNA analysis. Colons were sectioned such that proximal ends and distal ends of mucosa were placed into separate tubes for RNA profiling. We froze the remaining colon tissues in RNAlater at −80°C for RNA extraction. Blood was collected by cardiac puncture from mice from every group at termination. Approximately 250 μl of whole blood was collected into PAXgene tubes. Whole blood was assessed for RNA. Colons from all groups were measured for length and weight.

**Histological assessment**. Tissues were fixed in 10% buffered formalin, embedded in paraffin and cut into sections. Sections were stained with hematoxylin and eosin. Stained sections were examined for evidence of colitis using as criteria the presence of mononuclear inflammatory cells, erosions, glandular loss, elongation or distortion of crypts and hyperplasia.

**Murine colitis experiments (at Janssen/Boulder Biopath site): adoptive transfer**. 160 C57 RAG(−/−) mice, 80 C57Bl/6 RAG(−/−) and 80 C57Bl/6 from Taconic. Mice were 6–7 wks old C57Bl/6 were 12 weeks, and all mice were female. Mice were acclimated for at least 7 days after arrival. Mice were housed at 5 animals/cage. On Study day 0, C57Bl/6 mice were terminated and spleens obtained for CD45RBhigh and CD45RBlow cell isolation (Using IBD Cell Separation protocol). After cells were obtained and sorted, control animals (Group 1) received ~$4 \times 10^5$ CD45RBlow cells (100 μl/mouse injections) and disease animals (Group 2) received an IP injection of ~$4 \times 10^5$ CD45RBhigh cells (100 μl/mouse injections). On study day −1, mice were weighed and randomized into treatment groups (described below) based on body weight. 10 animals were randomly selected from each group and necropsied on study day 14, 21, 28, 35, 42, 49, 56, and 63. Intestinal tissue was flushed with cold PBS, mucosa removed and placed into separate tubes and flash frozen.

**Samples for RNAseq**.

 i. DSS colon: 3 timepoints (basal, 'middle', end), $n = 15$ mice = 45 samples
 ii. DSS blood: 3 timepoints, $n = 15$ mice (or pooling samples) = 45 samples
 iii. *Rbhi mucosa: 3 timepoints, $n = 10$ mice = 30 samples
 iv. *Rbhi muscularis: 3 timepoints, $n = 10$ mice = 30 samples
 v. Rbhi blood: 3 timepoints, $n = 10$ mice = 30 samples
 Total: 180 samples (90 samples if pooled two mice/analysis)
 Quality control further limited sequenced samples for analysis.

**RNAseq alignment**. Raw RNAseq reads were aligned against the mm10 reference genome using STAR (version 2.4.0g1)[21] and gene- and exon-level read summarizations were performed with the featureCounts program in subRead[22]. Ensembl annotations were used for both the alignment and the read summarization. For the exon-level read summarization, a collapsed version of the annotations file where gene designs are described based on non-overlapping "counting- bins" was used to avoid over- or under-counting of reads. The raw data files, the gene- and exon-level read count matrices, a sample descriptor file and an experimental procedure descriptor file have been uploaded to the Gene Expression Omnibus (GSE*).

**Differential expression and splicing at fixed timepoints**. The differential expression analysis was performed with the DESeq2 package in R[23].

**The differential splicing analysis was performed using the R limma package**. A voom transformation[24] was applied to the gene- and exon-level read count matrices and the resulting log2-transformed counts per million (log2cpm) matrices and the observation-level weights were fed to the lmFit function to fit the mixed linear model described in equation (1). The $p$ values associated with the DSS vs. control effect on each gene and exon were computed using empirical Bayes moderated $t$ statistics. The differential exon-usage estimations were done using the diffSplice function (limma) on the exon-level fit object and only those genes with

an FDR < 10% as given by either moderated F-statistics or simes-adjusted $t$ statistics were considered as differentially spliced.

**Differential expression and splicing trajectories**. For the identification of the genes that show different temporal trajectories of expression in the DSS and control groups, DESeq2 was used (11) to perform a likelihood-ratio test between a full model containing a disease-time interaction term fitted on gene-level count data $X$ and a reduced model without such term $Y$.

The identification of genes that show differential splicing trajectory across time in control and disease was done by fitting a model containing a disease-time interaction term on exon-level data and performing a simes-moderated $t$ test to obtain gene-wise significance of disease-time interaction using diffSplice (limma).

**Machine learning and validation models**. Each box in Fig. 4a reports on the performance of the model outlined in the flowchart of Supplementary Fig. 5. Random forest was used to solve

HistoScore ~ PC1_{gene} + PC2_{gene} + PC3_{gene}
+ PC1_{exon} + PC2_{exon} + PC3_{exon}
+ median_expression + median_splice + VDJ_{read_{cpm}} + VDJ_{clonality}

in the Janssen WC colonic dataset. This gives a function relating HistoScore to gene, exon usage, V(D)J burden and clonality, trained only on the 40 mice of the DSS Janssen cohort and only on the colonic expression signatures so derived. Here, PC1,2,3 are the first three principal components of the DSS WC signature for either genes or exon usage, median is the median expression (relative isoform expression) of the genes in that signature, and VDJ-read-cpm, VDJ-clonality are the adaptive inflammation scores for each mouse in that dataset, respectively. In order to validate the predictive power of the training function above, we use it to predict the HistoScore using each validation dataset. For each validation set of the $y$ axis we allow this training function to predict HistoScore. For each signature and each validation set we have a model with a predicted HistoScore, and in this model we can perform variable selection to pick which predictors are the most important to that model. Using k-fold cross validation, we avoid superflous predictors and generally mitigate overfitting. Also note that for each model, we keep the same number of predictors even though the number of genes in each signature can vary tremendously. In each box, we only report the top three predictors for that model.

**Differential expression and splicing calls**. The differential splicing of Il1rl1 and Lama3 was interpreted in terms of differential transcript expression by comparing the differential exon-usage pattern identified by diffSplice, the alignment pattern observed in the exploration of BAM files using the Integrative Genome Viewer (IGV)[25,26] and the transcript variants described in RefSeq[27] (both validated and predicted) for those genes.

**Human subnetwork generation**. To explore the relevance of the transcriptomic alterations identified in the DSS model for human disease, we used a colonic gene-regulatory network (GRN) built from microarray gene expression and genotype data from a population of Crohn's Disease (CD) patients[9]. The overlap between the murine differential expression signatures and the human GRN was done using Cytoscape 3.2.1[28].

**Experimental validation of differential transcript expression**. Experimental validation of the differential transcript usage for Il1rl1 and Lama3 was performed through a qPCR experiment using transcript- specific primers spanning exon junctions (Supplementary Data 5, 6). RNA (range 500 to 1000 ng) was reverse transcribed using the Super script IV Kit including an RNase H step (Invitrogen, 180901050). cDNA was diluted up to 1:4 prior to running the qPCR. Primers used for SYBR® green I detection (Power SYBR™ Green PCR Master Mix Applied Biosystems, 4368706) were Il1rl1 lacking TIR homology domain (Forward: AGGTCGAAATGAAAGTTCCAGC, Reverse: AGCAATGTGTGAGGGACACT), Il1rl1 with TIR homology domain (Forward: CGGAACGATGGCAAGCTCTA, Reverse: TGGATACTGCTTTCCACCACG), Lama3 lacking N-laminin (Forward: TCAGAGCAGCAAAGGGTAGC, Reverse: TGTGTTGTGCTGACAGTTAA TGC), Lama3 with N-laminin (Forward: CCTTGGATCTGGGTCAGCTCT, Reverse: CTGGGTAATTGCCATGTTTGCT). As control gene, we used expression of Mapre1 (Forward: TGATTTGCCAGGAGAACG, Reverse: GCCCCCTTCATC AGGTATCA) or Rplp0 (36B4; Forward: ATGGGTACAAGCGCGTCCTG, Reverse: GCCTTGACCTTTTCAGTAAG). The qPCR was run using the 7900HT Applied Biosystems Real-Time PCR System (Stage 1: 95 °C 2:00 (1 cycle); Stage 2: 95 °C 0:20, 55 °C 0:15, 72 °C 0:10 (40 cycles); Stage 3: dissociation stage). Data were analyzed using SDS 2.2.1 software.

**MSCCR cohort**. The Mount Sinai Crohn's and Colitis registry (MSCCR is a prospective cross-sectional cohort consisting of ~1500 IBD patients and controls recruited during their endoscopy visit from December 2013–September 2016. Paired blood and biopsy RNAseq transcriptome data collected alongside as well as histological, endoscopic and clinical assessments were determined at the time of their MSCCR endoscopy[29].

Biopsy RNAseq data were generated as previously described. Briefly biopsy and Blood RNA was extracted and processed in randomly allocated batchers as previously described[1]. RNA was isolated from frozen tissue using Qiagen QIAsymphony RNA Kit (cat.# 931636) on the QIAsymphony. RNA from whole blood collected in PAXgene tubes was isolated using QIAsymphony Blood PAXgene RNA kit (cat.# 762635). One microgram of total RNA was used for the preparation of the sequencing libraries using the RNA Tru Seq Kit (Illumina (Cat # RS-122-2001-48). Ribosomal RNA from biopsy tissue was depleted from total RNA using the Ribozero kit (Illumina Cat # MRZG12324), and globin RNA along with ribosomal RNA was depleted from total blood RNA using Globin zero gold rRNA removal kit (Illumina cat.# GZG1224) to enrich poly-adenylated coding RNA as well as non-coding RNA. The rRNA and globin + rRNA depleted RNA from biopsy and blood total RNA, respectively was used for preparation of the sequencing library using RNA Tru Seq Kit supplied by Illumina (Cat # 1004814). The ribozero and globin zero RNASeq libraries were sequenced on the Illumina HiSeq 2500 platform using 100 bp paired end protocol following manufacturer's procedure.

Genomic alignment to GRCh37 of single-end RNAseq reads was performed using 2-pass STAR[21]. Default parameters for STAR were used, as were those for the quantification of aligned reads to GRCh37.75 gene features via featureCounts[22]. Multimapping reads were flagged and discarded. Raw count data was pre-filtered to keep genes with CPM > 0.5 for at least 3% of the samples. After filtering, count data were normalized via the weighted trimmed mean of M-values[30].

Gene expression matrices were generated using the voom transformation and adjusted for technical variables (e.g. RIN, processing batch, rRNA rate, and exonic rate) using the *limma* framework. Expression matrices were also adjusted for age, gender, and genetic PCs for further analysis.

**Regulatory approval**. Icahn School of Medicine at Mount Sinai and Boulder Biopath received Institutional Animal Care and Use Committee (IACUC) approval for conducting these murine disease model experimental studies and have complied with all relevant ethical regulatory requirements. All human data were from previously published cohorts and therefore, IRB and informed consent was not required for this study.

**Appendices**. Authors submit individual source files to ensure readability. This file type is published in raw format and is not edited or composed.

**Statistics and reproducibility**. Statistical analysis was designed as described in the Methods section. All analysis was reproduced in at least two independent experiments.

**Reporting summary**. Further information on research design is available in the Nature Portfolio Reporting Summary linked to this article.

## Data availability

The MSCCR data and the blood and intestine expression data and clinical/demographic description of the MSCCR cohort including tables are described in: PMID: 32980345, PMID: 35190725, and PMID: 34780722. The MSCCR data are available on GEO (GEO accession: GSE186507 for blood and GSE193677 for biopsy). Data are private and will be released in October 2024 and 2025 (PMID: 36109152)[31]. Mouse model data are available at GEO accession: GSE214600: Go to. Numerical source data are provided here (Supplementary Data 10): https://doi.org/10.6084/m9.figshare.21706202.v2.

## Code availability

Code is provided in the supplementary data (Supplementary code 1). Open-source R code from publicly available packages that was exclusively used in this study is available here https://github.com/LosicLab/losiclab.github.io.

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

## Acknowledgements

B.L. acknowledges Eugene Fluder and HPC at ISMMS for valuable assistance and Genetics and Genomic Sciences at ISMMS for financial support. B.L. acknowledges key early efforts of Aritz Irizar. L.P., E.E.S., and C.A. were supported in part by The Leona M.

and Harry B. Helmsley Charitable Trust and RC2 DK122532/DK/NIDDK NIH and the Janssen/Mount Sinai Icahn Institute Collaboration in IBD.

## Author contributions

Conception and design were by A.D., B.L., L.P., C.A., J.R.M., M.C., R.D., E.S., J.F., H.A.R.; mouse experiments were designed and conducted by H.X., A.D., J.R.M., L.P., C.A., and H.A.R.; nucleic acid isolation and isoform validation was conducted by: J.H., T.D., M.P.A., C.A.; analysis and interpretation were by: B.L., L.P., C.A., S.H., J.R.F., A.S., P.R., J.R.M., A.D., H.A.R.; data curation and transfer: Z.C.; Drafting the manuscript for important intellectual content was managed by J.R.F., A.S., B.L., L.P., C.A.

## Competing interests

The authors declare the following competing interests: J.R.F., A.S., J.R.M., H.A.R., M.C., R.D. & A.D. are current or former employees of Janssen. J.R.F. is a current employee of Spark Therapeutics. J.R.M. is a current employee of Moderna. R.D. is a current employee of Pathos A.I. B.L. is a current employee of Guardant Health. The remaining authors declare no competing interests.
