## [Peer Review File · Communications Biology]

Reviewers' comments:

Reviewer #1 (Remarks to the Author):

In this study, the authors sought to characterize longitudinal changes in gene expression (bulk RNA seq), splicing patterns, and adaptive immune repertoire (VDJ sequences) in murine models of colitis (DSS and adoptive transfer) and associate these signatures with histologic inflammation in mice and in patients with inflammatory bowel disease (IBD). They identify genes whose expression/splicing patterns were relatively fixed (e.g. *Il1r1*, *Lama3*) or had dynamic temporal alterations (e.g. immunoglobulin genes) throughout the natural history of the murine colitis cycles. The authors identified gene modules/signatures that classified grades of histologic inflammation in mice and then integrated and translated these signatures derived from murine colitis to classify histologic inflammation in patients with IBD (using machine learning/random forest). Finally, the authors performed an integrated analysis of transcriptomic signatures from mice with gene expression/IBD network analysis in patients and identified shared acute (innate immune, IgA production) and more chronic (TCR signaling, antigen presentation) gene signatures that may reflect underlying IBD pathogenesis. The authors acknowledge that murine models of IBD may have limitations in recapitulating human IBD pathogenesis and thus sought to use a systems biology/bioinformatics approach to model and understand which transcriptomic/regulatory pathways may overlap in patients with IBD. Although prior studies have evaluated longitudinal gene expression patterns in murine models of IBD, this study is relatively unique in integrating gene expression with splicing information and VDJ signatures. This is an interesting study that highlights the importance of temporal variation in gene expression in murine colitis and extend this to patients with IBD. Furthermore, their AI-based prediction models for histologic inflammation could be refined to model human IBD pathogenesis with mice models. I have the following critiques and recommendations:

1. Results: Colitis in the mouse induces expression of minor *Il1r1* and *Lama3* isoforms: "The results of the pathway enrichment analysis performed on the genes positively correlated (Pearson's $R > 0.7$) with either *Lama3* transcript in RNAseq data show a shift towards immunity-related pathways for the genes correlated with the short transcript (Figure 3H)." Recommend clarifying in text which immune pathways were enriched to improve clarity for readers.
2. Methods/Results: The authors should provide more details on how they carried out their VDJ sequencing from blood and tissue in the murine colitis models. Were B and T cell enriched before immune repertoire sequencing? Could authors report TCR and BCR metrics (e.g. *TCRa*, *TCRB* genes, IG heavy and light chain genes) separately according to murine colitis cycle? It would also be helpful to plot changes in immune repertoire diversity from control and the different cycles of DSS and rest.
3. Results: "Figure 4 DSS temporal signature and trajectory predictive power of colonic histopathology across tissue, location, and colitis model" is rather dense and difficult to follow. While it is great that the authors provide Spearman coefficients and p-values, it is unclear what features/genes are predictive for histologic inflammation based on T cell or tissue source (blood or colon location). For example, under T cell 1 and distal mucosa, Pearson coefficient is 0.7 and appears to be a strong predictor. However, what does this mean? Is this total T cell infiltration in distal mucosa? Was this based on IHC staining (if so which markers)? Or was this based on bulk-RNA seq data? Was deconvolution and proportion of specific T cells quantified to derive this predictive signature?
4. Results: "Figure 5 Mouse temporal signatures are predictive of histology state in a cohort of inflammatory bowel disease patients" The author's primary outcomes were histologic inflammation measured by various scoring indices (Nancy Index, GHAS) which were then correlated with gene signatures (temporal gene signatures A, B, C, D, etc). It's not entirely clear which genes were part of which temporal signatures and the effect size/weights of individual genes in these signatures in predicting their outcomes. The authors should make this more clear for readers (include a new figure/table clearly delineating the composition of these gene signatures).

5. Results, Figure 5: It's interesting that neutrophils (PMNs) in epithelium was used as an outcome for histologic inflammation in the authors' predictive model. Were there any neutrophil-specific genes (e.g. VNN2, S100A8, S100A9, MPO, etc) from bulk RNA-seq that were predictive for histologic inflammation?

6. Results: It appears most of the data was modeled using DSS colitis, although the authors mentioned they also included adoptive T cell transfer models of murine IBD. Were there any immune repertoire changes comparing DSS colitis versus adoptive transfer models?

7. Results: Were the VDJ signatures also validated in patients with IBD?

8. Results: The authors argue that including information on differential splicing patterns was a novel aspect of this study. The clinical significance of evaluating this additional layer of genetic data is unclear. Were any of the differential splicing patterns observed in DSS colitis predictive of histologic inflammation?

9. Results/Discussion: The authors plotted differences in VDJ clonotypes and burden among different DSS colitis stages (Supplementary Figure 4). However, it is unclear if these immune repertoire metrics were evaluated in their predictive models for histologic inflammation. Was immune repertoire diversity correlated with histologic inflammation? Were there specific TCR or BCR clonotypes that were enriched in inflamed versus non-inflamed colonic tissue in the mice models?

10. Results: Testing the predictive power of expression and splicing signatures in IBD: "Taking the max score per individual, multi intestinal biopsies were averaged across all intestinal regions per individual in an independent cohort of 1000 IBD patients." The authors provide little to no information about the IBD patient cohorts they used to validate their gene signatures. The authors should provide a supplementary table with baseline clinical characteristics of included patients. Given that the gene signatures were derived from murine models of colitis (no small bowel inflammation), did they exclude patients with Crohn's disease with only small bowel involvement? Gene expression and immune signatures vary between small bowel (e.g. ileitis) and large bowel (colitis). It would not be appropriate to extrapolate the findings of this study with murine colitis to model small bowel Crohn's disease.

11. Results/Discussion: The authors should perform sensitivity analyses comparing the utility of their DSS-colitis-derived transcriptomic signatures in predicting histologic inflammation in patients with ulcerative colitis versus Crohn's colitis separately. Do their murine colitis-derived gene expression signatures recapitulate/resemble ulcerative colitis versus Crohn's colitis better?

12. Results/Discussion: Given that one of the goals of this study was to use DSS colitis-derived temporal transcriptomic signatures as biomarkers to classify or predict histologic inflammation in mice and patients with IBD, it would be appropriate to demonstrate the robustness of their models/various gene signatures by including receiver operating characteristic (ROC) curve analyses and calculating area under the curve (AUC) as well as sensitivity and specificity.

Reviewer #2 (Remarks to the Author):

Authors described a well thought experiment where colitis was induced in mice in a case-control setting. Authors collected samples at multiple timepoints and performed RNAseq as well as histopathology on the same specimens. This manuscript describes transcriptomics analysis comparing expression data between DSS treated and control mice at 5, 12, 17 and 36 days. Transcriptomics analyses were focused at the gene and transcript level discovering DEGs and differential splicing. Authors also clustered expression data to identify five distinct groups of genes with different expression trajectories.

Subsequently a set of random forest classifiers were used to predict histology scoring. While I appreciate the novelty of the analysis and the uniqueness of the dataset, there are several points that require clarification or further analyses.

When the authors discuss the two selected differentially spliced genes, they illustrate the rationale for selecting them but, ultimately, they do not provide results of that selection. What is the p-value and in how many timepoints those two genes differentially spliced?

Not enough details were provided on how RNAseq data was generated: which sequencer was used? How long were the reads? Was RNAseq data QC'd? Paired end or single end? Stranded or non stranded?

I strongly suggest performing a variance partition analysis (<https://bmcbioinformatics.biomedcentral.com/articles/10.1186/s12859-016-1323-z> Hoffman et al.) of RNAseq data to show how much of the variance is explained by variables such as time, RIN, weight and other available metadata. This would also help identifying covariates to adjust for when performing a differential analysis.

The description of the machine learning approach results confusing and there is no clear indication on which data is used, sample size and feature engineering (if any). Please, clarify how and from where the validation set was obtained. Is the feature selection performed on the whole dataset, validation set included? Was the feature selection nested within the 10-fold CV? Supplementary figure 5 should address this point but fails to deliver. Moreover figure S5 seems to be still a draft, and if in its final form, should be extensively revised.

Overall methods lacks in details and are still populated with author's comments. E.g. "... qPCR experiment using transcript- specific 531 primers spanning exon junctions (Table S1). (Additional details on the qPCR, Carmen?, SYBR Green was 532 used, right?). RNA (range 500 to 1000 ng) was reverse transcribed ... " and multiple question marks (?) here and there throughout the manuscript. Manuscripts should be thoroughly examined before submission.

POINT-BY-POINT RESPONSE

We appreciate the comments from the reviewers and thank them for their positive feedback. We believe that the revisions have strengthened our manuscript. Our detailed point-by-point responses to the comments are attached below. For clarity, we have established three writing modes in this document: **black-bold for the original editor's or reviewer's comment as included in the decision letter**, black-regular for our responses, and **red for the changes made to the original version of the manuscript**.

Reviewer: 1

1. Results: Colitis in the mouse induces expression of minor *Il1r1* and *Lama3* isoforms:

"The results of the pathway enrichment analysis performed on the genes positively correlated (Pearson's $R > 0.7$) with either *Lama3* transcript in RNAseq data, show a shift towards immunity-related pathways for the genes correlated with the short transcript (Figure 3H)." Recommend clarifying in text which immune pathways were enriched to improve clarity for readers.

Comment added clarifying the specific IL-5 signaling pathway enrichment on p. 5.

The results of the pathway enrichment analysis performed on the genes positively correlated (Pearson's $R > 0.7$) with either *Lama3* transcript in RNAseq data show a shift towards immunity-related pathways, **including IL-5 and Jak-STAT signaling** for the genes correlated with the short transcript (Figure 3H).

2. Methods/Results: The authors should provide more details on how they carried out their VDJ sequencing from blood and tissue in the murine colitis models. Were B and T cells enriched before immune repertoire sequencing? Could authors report TCR and BCR metrics (e.g. *TCRa*, *TCRB* genes, IG heavy and light chain genes) separately according to murine colitis cycle? It would also be helpful to plot changes in immune repertoire diversity from control and the different cycles of DSS and rest.

The reviewer raises a key point. We did not perform any special enrichment for B or T cells prior to immune repertoire sequencing, but instead used the MiXCR package to quantify B and T cell receptor RNA-seq reads mapping to VDJ loci and normalizing by total library size. In the third paragraph of page 6, we emphasize that our non-targeted-amplicon sampling of VDJ loci can only capture the leading order clonal structure of the adaptive response. Indeed formally the overall VDJ detection sensitivity for at least one clone at the sample level was 178/179 (99.4%) for the DSS model and 99/224 (44.2%) for the average adoptive T cell model. However, the median number of distinct clones detected per sample (a surrogate of VDJ signal strength) was 498 for DSS models and 8 for the adoptive T cell

models. This indicates that the detections in the adoptive transfer models are fairly sparse and probably within the noise band, while the average DSS model yields a modest but reasonable median profile of about 500 VDJ clones per sample (minimum 22, maximum 2969) (see also Supp Fig S1 a).

In terms of the concrete metrics requested by the reviewer, we did compute not only the V, J and in many cases D genes where possible including their start and end sites as well as the CDR3 aa & nucleotide sequences. These are now attached as a Supplementary Table. Our feeling is that these results, while suggestive, await targeted amplicon deep-sequencing for validation before more detailed inference can be claimed on specific V or D gene usage by disease cycle. Nevertheless, further to the last question from the reviewer, as shown in Supp Fig A b, we do conclude that diversity does trend upwards in later DSS cycles (especially in day 36, second rest)

We did observe that VDJ clones were detected in >99% of the DSS models at a median of 500 clones per sample, while on average 44% of samples in the adoptive T cell models had detections of only about 8 clones per sample.

3. Results: "Figure 4 DSS temporal signature and trajectory predictive power of colonic histopathology across tissue, location, and colitis model" is rather dense and difficult to follow. While it is great that the authors provide Spearman coefficients and p-values, it is unclear what features/genes are predictive for histologic inflammation based on T cell or tissue source (blood or colon location). For example, under T cell 1 and distal mucosa, Pearson coefficient is 0.7 and appears to be a strong predictor. However, what does this mean? Is this total T cell infiltration in distal mucosa? Was this based on IHC staining (if so which markers)? Or was this based on bulk-RNA seq data? Was deconvolution and proportion of specific T cells quantified to derive this predictive signature?

Yes we agree with the reviewer that Figure 4 is quite dense. Before diving into explaining it, let us just clarify that the only immune features we are using in these predictive models are derived from the RNA-seq data itself using either expression profiling or adaptive VDJ reconstruction as detailed in the previous question, ie there is no IHC staining information being used.

Concerning Figure 4, we are attempting to provide the top predictive DSS splicing and expression features and model performance across DSS cycles for validation sets encompassing different tissues (blood, proximal colon, distal mucosa) and even models and experimental sites (DSS cohorts). The top 3 predictive features of each model are printed on the bar plot, ranked by feature by importance (computed via mean decrease accuracy on OOB samples). The columns of the figure refer to what signature is being used in the feature list, whose principal components form the features of the model being tested. In summary, each cell in the figure includes not only the spearman coefficients and p

values but also the top 3 predictive features ranked in order of feature importance. Taking the reviewer's example, let us consider adoptive transfer T cell model 1 and distal mucosa with strong positive prediction of histopathology at day 36. The model is claiming that the second principal component of the DSS specific differential splicing signature at day 36 is strongly predictive of histopathological distal mucosa splicing patterns in the adoptive transfer T cell model 1 colitis model. In other words, if we determine which genes have differential exon usage between DSS whole colon and control mice whole colon at day 36, then those same genes have differential splicing patterns in distal mucosa tissue whose second principal component of variance is a strongly predictive feature of histopathological scoring in the adaptive T cell transfer model 1 colitis model. The second most important feature is the median gene score, which is just the average gene expression across the differentially expressed genes at day 36 in the DSS model. Taken together, this cell in Figure 4 indicates that DSS specific exon usage at day 36 in the DSS model is capturing the same usage profile in a completely different tissue for a completely different model of colitis, suggesting a common underlying mechanism highlighting splicing. Note that if we look at the adjacent cell and combine all DSS cycles together ("all days") and look at average expression and splicing patterns (instead of just those at day 36) we see a significant *negative* correlation to histopathological scoring in distal mucosa. Scanning this row we see that temporal cluster D genes are also negatively correlated with histoscore. Taken together these observations suggest that cluster D genes specifically codify an underlying temporal which may drive this average negative (healing) correlation with histoscore. However, caution should be taken in interpreting the adoptive transfer histopathology correlations because the dynamic range of their histopathology scores is low as shown in Figure 5B.

We agree with the reviewer that we could have been more clear in explaining the nature of Figure 4, so we have attempted to clarify this in detail by inserting some introductory discussion around Figure 4 on page 7:

“ For each signature, we constructed a random forest model using the same DSS whole-colon training set to predict histological scoring that incorporated predictive terms related to the specific fixed and trajectory expression, splicing, and adaptive VDJ signatures evaluated on the validation data (see Methods). Thus, as shown in Figure 4, we tested the performance, and feature importance, of multiple signatures on multiple validation sets.”

4. Results: "Figure 5 Mouse temporal signatures are predictive of histology state in a cohort of inflammatory bowel disease patients" The author's primary outcomes were histologic inflammation measured by various scoring indices (Nancy Index, GHAS) which were then correlated with gene signatures (temporal gene signatures A, B, C, D, etc). It's not entirely clear which genes were part of which temporal signatures and the effect size/weights of individual genes in these signatures in predicting their outcomes. The authors should make this more clear for readers (include a new figure/table clearly delineating the composition of these gene signatures).

We agree with the reviewer that this information could be presented more clearly. On page 4 we have explicitly highlighted that our DE gene signatures for fixed DSS timepoints are given in Supplementary Table 2. In the process we noticed we omitted including the analogue of this table for our 725 DSS cycling genes, so we have included those along with median normalized expression for each DSS cycle day (5,12,17,36) across control and disease mice in order to better facilitate downstream usage in addition to exploring functional significance of each gene. We have also highlighted that, as in the DSS prediction models whose performance is depicted in Figure 4, each homologous signature was decomposed into the first several principal components (PC1 - PC5) and only these principal components were used as features in the models for the human validation cohort.

New first paragraph of page 8:

The murine model molecular signatures were then tested for their ability to predict the Nancy (UC) and GHAS (CD) histological scores in human IBD patient biopsies (Figure 5). Taking the max score per individual, multiple intestinal biopsies were averaged across all intestinal regions per individual in an independent cohort of 1000 IBD patients. We show how the performance in predictive power for each phenotype is recapitulated using the homologs of the colitis evolution signatures derived in the DSS model. **More specifically, exactly as in the DSS models, each signature was broken down into the first several principal components (PC1 - PC4) and these were used in the prediction models.**

5. Results, Figure 5: It's interesting that neutrophils (PMNs) in epithelium was used as an outcome for histologic inflammation in the authors' predictive model. Were there any neutrophil-specific genes (e.g. VNN2, S100A8, S100A9, MPO, etc) from bulk RNA-seq that were predictive for histologic inflammation?

We thank the reviewer for bringing up this interesting point – yes, we did generally find that S100A8 (calprotectin) figured prominently in signature A. Similarly, in the DSS model we found that S100b was classified in cluster C. We did attempt to use individual genes in simpler models to assess performance however none of these simpler “singleton” models achieved the same predictive power we describe here in this work.

6. Results: It appears most of the data was modeled using DSS colitis, although the authors mentioned they also included adoptive T cell transfer models of murine IBD. Were there any immune repertoire changes comparing DSS colitis versus adoptive transfer models?

Yes we only used a single cohort of DSS colitis data to train our predictive models, and left the rest for holdout validation (shown in Figure 5) . In Supp Figure 4 – as well as in our new discussion on page 4 – we emphasize and show that the adaptive immune repertoire in the adoptive transfer model data is very sparse and barely above the detection threshold, as expected given the relative paucity of T cells from T cells injected into a RAG-/-.

7. Results: Were the VDJ signatures also validated in patients with IBD?

The reviewer brings up an interesting question. We have not yet directly recapitulated the low-pass RNA-seq derived VDJ landscape in IBD patients and we feel this is out of scope for the present study.

8. Results: The authors argue that including information on differential splicing patterns was a novel aspect of this study. The clinical significance of evaluating this additional layer of genetic data is unclear. Were any of the differential splicing patterns observed in DSS colitis predictive of histologic inflammation?

The referee raises a great point here. Splicing patterns are typically difficult to assign clinical significance outside well-studied special cases (for example isoforms of the insulin receptor coding genes, or those of BRCA) where the included or omitted exonic domain codes for a well understood corresponding protein domain. The functional or even clinical significance of complex patterns of splicing is generally difficult, if not impossible, to assess based on a detailed functional knowledge of constituent isoform-specific expression. Nevertheless this is an important mechanism of potentially generating disease specific proteomic diversity and even an approximate method may be able to capture a distinct, average, splicing effect distinct from that of gene expression. In this paper we first identify differential splicing that is DSS specific, highlighting the functional significance in *Il1rl1* and *Lama3* but also finding over a hundred others that are co-spliced in a DSS specific fashion. We find that these differentially spliced genetic loci not themselves differentially expressed, potentially representing an orthogonal source of signaling characterizing the DSS state. In order to assess if the principal components of the gene splicing signal are important features relative to VDJ and gene expression principal components, we allowed each model in Figure 4 “the same opportunity” to select splicing, expression, or adaptive immune features as most important to predict a particular phenotype. From Figure 4, we see that the model based on distal mucosa expression measured in the adoptive transfer model found that the second largest principal component of the splicing signature of the day 36 DSS model was the most important feature to predict histological scoring (spearman ~ 0.7 , $-\log_{10}(p) \sim 4.9$). This indicates that a “DSS rest” splicing signature is implicated in histological disease scoring in this context.

9. Results/Discussion: The authors plotted differences in VDJ clonotypes and burden among different DSS colitis stages (Supplementary Figure 4). However, it is unclear if these immune repertoire metrics were evaluated in their predictive models for histologic inflammation. Was immune repertoire diversity correlated with histologic inflammation? Were there specific TCR or BCR clonotypes that were enriched in inflamed versus non-inflamed colonic tissue in the mice models?

The immune plots of Supp Fig 4 are based entirely on the VDJ profiling we describe in response to the reviewer's second question above. We use these same VDJ measures (normalized VDJ expression and reads per clone) in each model trained and evaluated in Figure 4. Histological inflammation was correlated with these measures in Supp Fig 7, where it is shown they are well correlated. These molecular measures of inflammation are thus well-suited as features in predictive models. In terms of searching for specific clones dominating the most inflamed states, because of the non-targeted nature of these random-primer VDJ mappings, we decided it would be too speculative to speculate on common sequences.

10. Results: Testing the predictive power of expression and splicing signatures in IBD: "Taking the max score per individual, multi intestinal biopsies were averaged across all intestinal regions per individual in an independent cohort of 1000 IBD patients." The authors provide little to no information about the IBD patient cohorts they used to validate their gene signatures. The authors should provide a supplementary table with baseline clinical characteristics of included patients. Given that the gene signatures were derived from murine models of colitis (not small bowel inflammation), did they exclude patients with Crohn's disease with only small bowel involvement? Gene expression and immune signatures vary between small bowel (e.g. ileitis) and large bowel (colitis). It would not be appropriate to extrapolate the findings of this study with murine colitis to model small bowel Crohn's disease.

We now include a Supplementary Table, to describe the demographics of the MSCCR cohort. We did not specifically exclude ileal only CD, but took the max histology score across regions, appreciating there can be sub-clinical molecular inflammation in various intestinal regions of IBD patients.

11. Results/Discussion: The authors should perform sensitivity analyses comparing the utility of their DSS-colitis-derived transcriptomic signatures in predicting histologic inflammation in patients with ulcerative colitis versus Crohn's colitis separately. Do their murine colitis-derived gene expression signatures recapitulate/resemble ulcerative colitis versus Crohn's colitis better?

Crohn's can affect both the ileum and/or the colon, whereas ulcerative colitis only has colonic involvement. It is generally accepted in IBD research that the DSS murine model tends to recapitulate ulcerative colitis more effectively than Crohn's because of its chemical abrasion to the intestinal barrier. However in chronic colitis, it may resemble more colonic Crohn's. The adoptive transfer model tends to cause more disease in the proximal intestine closer to the small intestine and due to this regional manifestation it may better model ileal Crohn's disease. Due to these attributes, it would not be particularly informative to divide the prediction between Crohn's and ulcerative colitis. Thus these attributes do not warrant additional analysis for this study.

12. Results/Discussion: Given that one of the goals of this study was to use DSS colitis-derived temporal transcriptomic signatures as biomarkers to classify or predict histologic inflammation in mice and patients with IBD, it would be appropriate to demonstrate the robustness of their models/various gene signatures by including receive operating characteristic (ROC) curve analyses and calculating area under the curve (AUC) as well as sensitivity and specificity.

The referee raises a good point. However, since we are regressing on a continuous variable, the spearman correlation or median absolute error of the histopathological score is the relevant model performance metric. ROC curves are useful for classification only. We have inserted a clarifying remark on the expanded explanation (within Point 3 above) of at the bottom of page 6.

For each signature, we constructed a random forest model using the same DSS whole-colon training set to predict histological scoring that incorporated predictive terms related to the specific fixed and trajectory expression, splicing, and adaptive VDJ signatures evaluated on the validation data (see Methods). Thus, as shown in Figure 4, we tested the performance, and feature importance, of multiple signatures on multiple validation sets. **Since our objective variable, histological scoring, is continuous we assess performance by spearman correlation and its associated p-value.**

Reviewer: 2

Authors described a well thought experiment where colitis was induced in mice in a case-control setting. Authors collected samples at multiple timepoints and performed RNAseq as well as histopathology on the same specimens. This manuscript describes transcriptomics analysis comparing expression data between DSS treated and control mice at 5, 12, 17 and 36 days. Transcriptomics analyses were focused at the gene and transcript level discovering DEGs and differential splicing. Authors also clustered expression data to identify five distinct groups of genes with different expression trajectories.

Subsequently a set of random forest classifiers were used to predict histology scoring. While I appreciate the novelty of the analysis and the uniqueness of the dataset, there are several points that require clarification or further analyses.

When the authors discuss the two selected differentially spliced genes, they illustrate the rationale for selecting them but, ultimately, they do not provide results of that selection. What is the p-value and in how many timepoints those two genes differentially spliced?

Not enough details were provided on how RNAseq data was generated: which sequencer was used? How long were the reads? Was RNAseq data QC'd? Paired end or single end? Stranded or non stranded?

I strongly suggest performing a variance partition analysis (<https://bmcbioinformatics.biomedcentral.com/articles/10.1186/s12859-016-1323-z> Hoffman et al.) of RNAseq data to show how much of the variance is explained by variables such as time, RIN, weight and other available metadata. This would also help identifying covariates to adjust for when performing a differential analysis.

The description of the machine learning approach results confusing and there is no clear indication on which data is used, sample size and feature engineering (if any). Please, clarify how and from where the validation set was obtained. Is the feature selection performed on the whole dataset, validation set included? Was the feature selection nested within the 10-fold CV? Supplementary figure 5 should address this point but fails to deliver. Moreover figure S5 seems to be still a draft, and if in its final form, should be extensively revised.

Overall methods lacks in details and are still populated with author's comments. E.g. "... qPCR experiment using transcript- specific 531 primers spanning exon junctions (Table S1). (Additional details on the qPCR, Carmen?, SYBR Green was 532 used, right?). RNA (range 500 to 1000 ng)

was reverse transcribed ... “ and multiple question marks (?) here and there throughout the manuscript. Manuscripts should be thoroughly examined before submission.

We thank the reviewer for these great questions and address them one by one in the following:

When the authors discuss the two selected differentially spliced genes, they illustrate the rationale for selecting them but, ultimately, they do not provide results of that selection. What is the p-value and in how many timepoints those two genes differentially spliced?

We thank the referee for pointing out this omission to us. The early DSS specific Lama3 isoform has a FDR of 2.39e-18 (row 1 of Table S2), while the early DSS specific I1r11 isoform has an FDR of 1.4e-e3. Both are differentially spliced on day 5 (first hit DSS cycle), while Lama3 is also differentially spliced on day 36. We have now included the following text:

Among the 246 genes that showed significant evidence of being differentially spliced at any of the timepoints (Figure S1) we identified the genes that a) have an FDR < 10⁻⁴ for differential splicing at least one timepoint and b) whose differential exon usage pattern can be explained by differential expression of annotated transcripts (in Refseq and/or Ensembl). Two genes met those criteria: I1r11 (ST2), a receptor in both membrane bound and soluble forms belonging to the Toll-like receptor superfamily whose ligand is I1-33, and Lama3, a secreted protein that belongs to the laminin family and acts as the alpha subunit of laminin-5 heterotrimer.

The early DSS specific Lama3 isoform has a FDR of 2.39e-18 (row 1 of Table S2), while the early DSS specific I1r11 isoform has an FDR of 1.4e-e3. Both are differentially spliced on day 5 (first hit DSS cycle), while Lama3 is also differentially spliced on day 36.

Not enough details were provided on how RNAseq data was generated: which sequencer was used? How long were the reads? Was RNAseq data QC'd? Paired end or single end? Stranded or non stranded?

As outlined in the Materials and Methods section the Illumina HiSeq 2500 instrument was used to generate unstranded, single-end 76 bp reads. We used fastQC to perform quality assessment.

I strongly suggest performing a variance partition analysis (<https://bmcbioinformatics.biomedcentral.com/articles/10.1186/s12859-016-1323-z> Hoffman et al.) of RNAseq data to show how much of the variance is explained by variables such as time, RIN, weight and other available metadata. This would also help identifying covariates to adjust for when performing a differential analysis.

We agree with the referee. We actually already carried out a variance partition analysis to parse out the variance contributions of three separate models, using random effects to model for categorical variables, but did not include it in the original manuscript. Here is what we did:

model 1: effect of disease (ctrl_disease), day (hit/rest cycle), interaction of day and disease, RIN and total RNA extracted per mouse

```
form <- ~(1|ctrl_disease) + (1|day) + (1|day:ctrl_disease) + RIN + total_RNA_ug
```

model 2: same as model 1, except day is redefined to be fixed effect variable with equidistant values [1 (hit),2 (rest),3 (hit),4 (rest)] (day2)

```
form <- ~(1|ctrl_disease) + day2 + (ctrl_disease:day2)+ RIN + total_RNA_ug
```

model 3: same as model 2, except day is redefined to be fixed effect variable with nominal experimental values [5 (hit), 12 (rest), 17 (hit), 35 (rest)] (day2)

```
form <- ~(1|ctrl_disease) + day2 + (ctrl_disease:day2)+ RIN + total_RNA_ug
```

Our rationale in computing the variance profile of these three models is threefold:

- a) establish that the DSS phenotypes, along with DSS cycle, induce more significant variation in gene expression than RIN (RNA quality) or total RNA yield per mouse;
- b) specifically ask if the interaction between the DSS state and DSS cycle (time) is inducing significant variance even when including the marginal effects of state and cycle with RIN and total RNA yield. This directly establishes that our ‘cycling gene’ expression profiles are well-posed;
- c) compute the specific effect of treating DSS cycle as a fixed effect with and without nominal units;

As shown in the newly added Supp Fig 6, we find that disease phenotype (ctrl_disease) or its interaction with disease cycle (day or day2) is the leading source of variation in the dataset in all three models. Interestingly, depending on whether or not ctrl_disease is treated as a fixed or random effect we obtain different effect sizes for both the interaction and disease phenotype effect. We believe it should be treated as a random effect in this case, and so model 1 was the model used in computing differential expression and splicing. We note that other covariates were strongly correlated with disease phenotype (eg weight, colon length), and so were not included in the model.

The description of the machine learning approach results confusing and there is no clear indication on which data is used, sample size and feature engineering (if any). Please, clarify how and from where the validation set was obtained. Is the feature selection performed on the whole dataset, validation set included? Was the feature selection nested within the 10-fold CV? Supplementary figure 5 should address this point but fails to deliver. Moreover figure S5 seems to be still a draft, and if in its final form, should be extensively revised.

We regret any confusion. There is one training set, based on the whole-colon tissue DSS experiment conducted at Janssen. The murine validation sets span across distinct CROs (Sinai, Janssen), regional tissue (proximal, distal mucosa, blood), and colitis induction model (adoptive transfer, DSS), as described in the Materials and Methods section starting on page 17. These are listed as rows in Figure 4. The human validation sets depicted in Figures 5 and 6 are extensively described under “MSCCR cohort” on page 21.

Model training was only performed on the whole-colon tissue DSS experiment (see also page 20, *Machine learning and validation models*), utilizing the principal components of the expression and splicing signatures along with VDJ terms under $n = 10$ cv. Validation performance is given on Figure 4. Supplementary Figure 5 has been revised.

REVIEWERS' COMMENTS:

Reviewer #2 (Remarks to the Author):

The authors have provided a robust response to my critiques.

Reviewer #3 (Remarks to the Author):

Authors addressed all the concerns raised in the first round of review. No further comments.